psychology

motor processing, mediolateral leaning, false-belief tracking, anticipatory looking, adults

**Author for correspondence:**
Giovanni Zani
e-mail: giovanni.zani@vuw.ac.nz

# Mindreading in the balance: adults' mediolateral leaning and anticipatory looking foretell others' action preparation in a false-belief interactive task

Giovanni Zani[1], Stephen A. Butterfill[2] and Jason Low[1]

[1]School of Psychology, Victoria University of Wellington, Wellington 6140, New Zealand
[2]Department of Philosophy, University of Warwick, Coventry CV4 7AL, UK

GZ, 0000-0003-4733-9290

Anticipatory looking on mindreading tasks can indicate our expectation of an agent's action. The challenge is that social situations are often more complex, involving instances where we need to track an agent's false belief to successfully identify the outcome to which an action is directed. If motor processes can guide how action goals are understood, it is conceivable—where that kind of goal ascription occurs in false-belief tasks—for motor representations to account for someone's belief-like state. Testing adults ($N = 42$) in a real-time interactive helping scenario, we discovered that participants' early mediolateral motor activity (leftwards–rightwards leaning on balance board) foreshadowed the agent's belief-based action preparation. These results suggest fast belief-tracking can modulate motor representations generated in the course of one's interaction with an agent. While adults' leaning, and anticipatory looking, revealed the contribution of fast false-belief tracking, participants did not correct the agent's mistake in their final helping action. These discoveries suggest that adults may not necessarily use another's belief during overt social interaction or find reflecting on another's belief as being normatively relevant to one's own choice of action. Our interactive task design offers a promising way to investigate how motor and mindreading processes may be various integrated.

## 1. Introduction

The ability to track others' goals—the outcomes to which people's actions are directed—is essential for social functioning, and allows

us to predict others' movements. In certain situations, however, successful action observation requires belief-tracking to inform goal ascription. Suppose, for example, that Sally is going to perform an action the goal of which is to grasp her wedding ring. She falsely believes that her ring is in the right-side box, whereas actually, it is inside the left-side box. If we ignore the agent's false belief (FB), then fixing only upon grasping the ring as the goal of Sally's action would generate the wrong expectation of how her action would unfold—incorrectly predicting that she would move to the left-side box to reach and grasp her ring [1]. In this case, we need to track Sally's belief to correctly identify the motor outcome of her action, which is that she would move to the empty right-side box to reach and grasp the ring. Little is known, however, about the functional relationship between rapid belief-tracking and motor processes for supporting action observation and understanding.

The study of action observation and understanding has been tackled from two distinct theoretical approaches. One approach emphasizes that human beings have a social–cognitive system that guides rapid anticipations of others' belief-based action [2,3], and this fast mindreading system may operate even when the tracking of belief-relevant information is immaterial to the task at hand [4–6]. Another approach emphasizes the pre-reflective role played by human beings' motor system in the understanding of others' actions. Here, the idea is that action observation elicits motor representations and processes in the brain of the observer similar to that which would occur if the observer were to plan and execute that act [7–9]. Exploitation of planning-like motor processes elicited during action observation will ensure that, often enough, outcomes represented motorically in the observer of an action are actually goals of the observed action. In this way, some forms of goal ascription could be achieved in a cognitively efficient way whereby the only representations are motor representations [10].

Mindreading and motor approaches to action observation and understanding are often considered separately. The separation might partly be the result of the different approaches' task concerns. In the mindreading field, experimental tasks are designed to focus on mental representations underpinning observers' tracking or reasoning about another person's mistaken belief-based behaviour [11]. While informative, such tasks tend to neglect potential activations in participants' own motor system that can be revelatory of observers' rapid online action understanding. The focus of research in the motor arena, by contrast, has been on situations where the action is going to be successfully achieved by the agent. As our example of Sally's search for her wedding ring indicates, however, there are also situations in which an agent will have an FB about which actions are needed. This creates a challenge for action observation: if we are to predict how someone's action will unfold, we cannot always rely on how things actually are but must also take into account what the agent believes. The challenge provides motivation for our current work to investigate whether belief-tracking could map onto or modulate motorically grounded expectations about the goals to which an agent's actions are likely to be directed.

Research indicates that when motor representations and their planning-like processes generate expectations about someone's goal, they take into account various facts about the agent's environment (e.g. location in space of observed motor acts, object visibility) [12,13]. For Butterfill & Apperly [1], the fact that such processes occur in motor imagery [14] suggests that planning-like processes are not tied to the actual environment but can also generate behavioural expectations based on non-actual environments. A case in point is that mental imagery of an action invoked by linguistic stimuli, for example, can activate motor plans and other action properties such as postural leaning that is similar to that implemented during execution of such movement. Zwaan et al. [15] found that when participants stood on a balance board with the instruction to read sentences implying a forward-leaning posture (e.g. 'He dove into the pool') or a backward-leaning one (e.g. 'The teenager plopped down on the couch'), readers' own postural sway was congruently influenced by the implied action. Research also suggests that the neural representation of motor imagery and action observation is similar to that of motor execution [16,17]. Against this background, there is at least coherence in Butterfill and Apperly's theoretical conjecture that when observing an agent, the observer's motor system may generate expectations by taking into account not only facts about the actual environment but also facts about the environment as specified by the agent's belief or belief-like state. There is some initial plausibility to the conjecture: van der Wel et al. [4] found that information about someone else's belief systematically perturbed the motor processes underpinning the trajectory of adults' own hand movements on a computer mouse-tracking task. Whereas van der Wel et al. measured interference effects, we sought to uncover evidence of whether adult observers' FB tracking could modulate the pre-reflective role played by the motor system in the course of participants' interaction with an agent.

We used an interactive helping task to service our investigation. In the classical version of the helping task [18], participants observed an agent store an object inside box A. In the FB condition, while the agent

was absent, an experimenter transferred the object into box B and then shut both boxes. In the true belief (TB) condition, the agent saw the experimenter move the object to box B. In both conditions, the agent approached the now-empty box (box A) and unsuccessfully tried to open it. The participants tested were 18-month-olds and 2.5-year-olds, and children's final helping action suggested some sensitivity to the agent's belief about the content of the boxes: children opened the now-full box (box B) for the agent in the FB condition, and they opened the now-empty box for the agent in the TB condition. (This interpretation has been challenged by Priewasser *et al.* [19] whose findings indicate that children's performance may be driven by tracking another's ignorance rather another's FB; our conclusion will not depend on which interpretation is correct.) We expanded the functionality of the task in two novel and theoretically grounded ways.

First, we measured adult observers' online belief-tracking in the helping task by outfitting participants with wearable eye-tracker glasses. Some studies contend that rapid tracking of others' beliefs can be under automated processes and can be reflected in anticipatory looking responses [5]. We should be careful to acknowledge, however, that eye movements can be controlled by multiple kinds of processes simultaneously [20], with contributions from offline cognitive control (you can, for instance, move your eyes in response to instructions). Our suggestion is merely that anticipatory eye movements—measured prior to the agent selecting a particular action—gives us a reasonable chance of picking up on FB tracking that may be underpinned by some cognitively efficient mindreading system. We predicted that in the FB condition of the helping task, just before the agent is about to select an action, observers would look in anticipation towards the empty box. Similarly, in the TB condition, just before the agent selects an action, observers would look in anticipation towards the full box.

Second, we measured adult observers' motor-generated behavioural expectations by having participants stand on a Wii™ balance board (WBB), to provide temporally and spatially sensitive information about rapid changes in distribution of the body's centre of pressure in an online manner. Studies show that the WBB can reliably detect how motor representations activated during action observation and processing can elicit corresponding structures in motor control, resulting in unintended behavioural changes in motor output such as postural adjustments [14,21–23]. Studies also suggest that imagined movements produce subliminal electromyographic activity in the involved muscles and may be evidenced through perturbations in postural sway [24,25]. Such postural adjustments have been considered to reflect autonomic preparation occurring downstream from central motor planning [26,27]. Leaning can provide a window into the unfolding of action prediction in observers' motor system, with leaning potentially being either a pre-reflective or spontaneous indicator of prediction generated at an early point.

With respect to our helping task set-up where the agent's goal was to retrieve a target object, shifts in participants' mediolateral balance (leftwards–rightwards leaning) were of theoretical importance. We predicted that shifts in participants' leaning—sampled at an early time point when there were no overt cues to suggest which box the agent would ultimately choose to open—would pick up on FB tracking modulating motorically grounded expectations of the agent's actions. We specifically predicted that observers would lean in the direction they anticipate the agent will go, given her belief about the object location; observers would lean towards the empty box in the FB condition and lean towards the full box in the TB condition. However, if leaning reflects motorically grounded expectations of the agent's action that are independent from FB tracking, then observers would lean in the direction of the box containing the target object; observers would lean towards the full box in both the TB and the FB conditions.

Finally, though, what about later indicators of action understanding, particularly adults' final helping action? In other words, will adults' ultimate choice of action (either opening up the empty box or the full box) be the same as those of children in a Buttelmann *et al.* [18] style of helping task? We do not think that this will necessarily be the case because there is no normatively correct helping response to the task [28]. This is indicated by failures to conceptually replicate the original findings of Buttelmann *et al.*; apparently irrelevant changes to the procedure, such as having children sit at a table, can affect whether a difference in the ultimate helping behaviour between the TB and FB conditions is observed [29]. Consequently, we did not attempt to make any predictions about adults' final helping action. We were, however, committed to viewing the helping task as having the untapped potential for documenting spontaneous motor processing in the understanding of others' actions. Consequently, our overarching prediction focused on the different indicators of early action understanding: we predicted that spontaneous leaning and gazing would overlap in response patterning, with both metrics foreshadowing prediction of the agent's action rather than the observer's action.

# 2. Method

## 2.1. Participants

Buttelmann *et al.*'s [18] interactive task, focused on differences in participants' final helping action, indicated that human beings from as young as 18 months of age respond differently when others have a TB or an FB about an object's location. Buttelmann *et al.*'s findings suggested that the interactive task could be suitably used with older participants; the task's effect size relating to, for example, 2.5-year-olds' final helping action ($X(1, N = 24) = 8.22$, $p = 0.004$), was a large one at $W = 0.585$ (the square root of the $\chi^2$ statistic divided by the sample size). Several recent studies have found that the differences in the ultimate helping behaviour between TB and FB conditions are difficult to replicate. Given the recent challenges over replication efforts, it was important to have power of 0.90 (rather than the conventional 0.80) to minimize the risk of failing to find any apparently real effect of differences in final helping action between the TB and FB conditions. G*Power 3.1 [30] indicated that a sample size of 31 would be needed to reach the desired power of 0.90 for the Buttelmann *et al.* style of data characteristics (input: $W = 0.585$, error probability = 0.05; d.f. = 1). It has been noted by researchers [29] that the Buttelmann *et al.* effect size was qualified by a high data exclusion rate; 54% of the total collected sample was eliminated for reasons such as participants' lack of cooperation, participants' refusal to touch or help open any box, or experimenter error. Being cautious to accommodate for a potentially high data exclusion rate as suggested by the original study would result in recruiting at least 16 more participants ($0.54 \times 31$) to G*Power's sample size calculation, resulting in a minimum total sample of 47 participants. A total of 50 undergraduate students volunteered to take part in the current study that was advertised as an experiment on action observation. All participants were individually tested in a single-trial session. The data from eight adults were excluded (16%) from formal data analyses due to actor error (3), experimenter error (2) or participants failing attention-memory checks in the post-experiment questionnaire (explained in the Procedure section) (3). Of these excluded participants, 3 had been assigned to the FB condition and 5 had been assigned to the TB condition. The final sample comprised 42 adults (24 assigned to TB condition, 18 assigned to FB condition; $M = 19.2$ years, range = 18–25 years, 32 females).

## 2.2. Procedure

The experimental room was prepared before each participant's arrival. A table was placed in the centre of the room. On the top of the table, there were two boxes (each 10 cm high, 5 cm wide) and a pair of soundproof headphones connected to a mobile phone. The two boxes had a hidden unlocking mechanism: in order to open the lid, a push force had to be exerted on two specific corners of the top surface (figure 1). A fully functional WBB was placed 350 cm from the boxes and faced the front side of the table. A mock Wii™ Balance Board (mWBB) was positioned to face the opposite side of the table, 50 cm from the boxes.

After the participant signed the consent form, the experimenter indicated that a second participant was about to arrive (the second person was a confederate). In the meantime, the experimenter demonstrated how the boxes could be closed and opened. When the participant had successfully closed and opened the boxes, he or she was brought to stand on the platform (which was the fully functional WBB) facing the table. The experimenter helped the participant to wear the Tobii Pro Glasses 2, and then the experimenter performed Tobii Pro Glasses and WBB calibration (each device acquired information at 50 Hz sampling rate). The WBB was connected to a computer via Bluetooth and custom software (provided by Nathan van der Stoep at https://www.multisensoryspacelab.com/) was used to record body leaning. Gaze was recorded with Tobii Pro Lab. Synchronous recording of WBB and Tobii Pro Glasses data was managed by AutoHotkey v. 1.1.29.01.

As soon as the calibration procedure had been completed, the confederate knocked at the door. In order to strengthen the participant's conviction that the other person was a second participant, the experimenter showed the confederate where to leave her belongings, invited her to fill and sign the consent form, and then instructed her to stand on the platform (which was the mWBB) facing the other side of the table, and to wait until its calibration was completed.

While the confederate and the participant stood facing each other on their respective balance boards, the experimenter positioned himself between them (being careful to ensure that the participant's line of sight to both boxes was maintained). The experimenter's instructions to the confederate and the participant were as follows. 'We are about to begin the experiment. Confederate (name used), I will

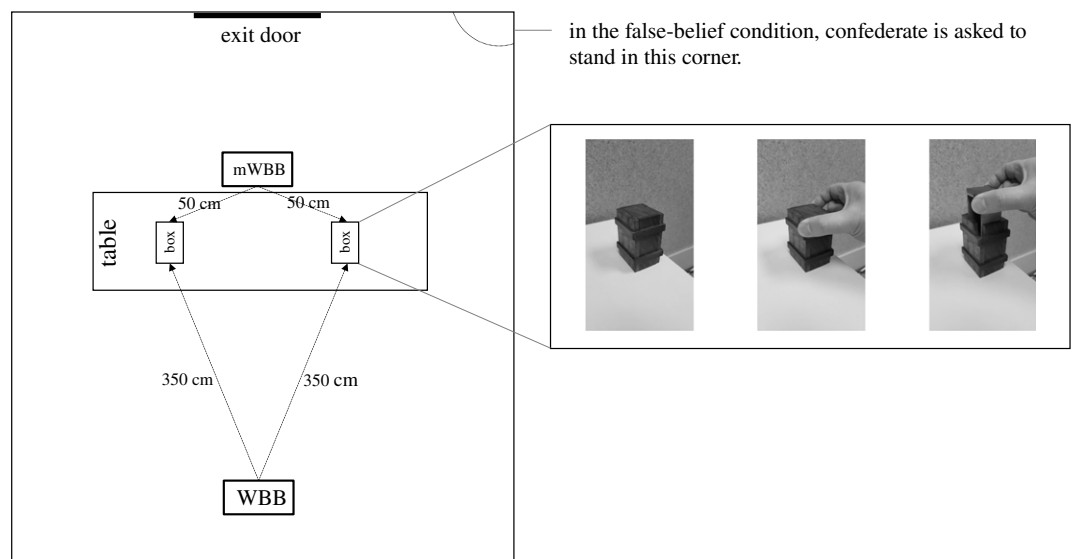

**Figure 1.** Schematic of the experimental setting and the boxes' unlocking mechanism.

ask you to perform some actions. Your task is to complete them but, if you need help, you can ask the other participant to come help you. Participant (name used), your task is to observe and, if the other participant asks for help, you have to go help her. Now we can begin the experiment. Confederate, do you have a small personal item such as a coin or a ring with you?' The confederate was trained to say, 'Is this ring ok?' A scripted conversation then took place: the experimenter said, 'It is perfect, can you put that ring in one of the two boxes?'; the confederate replied, 'Either one?'; and the experimenter stated, 'Yes'. At this point, the confederate put the ring in one of the two boxes (the initial location of the ring was counterbalanced throughout the experiment). The versions of what happened next differed according to the experimental condition. Each participant was assigned to either the FB condition or to the TB condition.

For participants in the FB condition, the experimenter spoke aloud to the confederate as follows. 'Now I am going to ask you to put the headphones on and to turn around towards this corner' (the experimenter showed the confederate that she would have had to step off the platform and to go to a corner of the room behind her, from which she could not see the table). 'This is the sound you will be hearing on the headphones' (he played a loud white noise that could be heard clearly by the participant). 'I will then leave the room and, as soon as you are ready, you have to close the door behind me and then you have to retrieve your object. Ok, now put the headphones on and go to the corner'. Once the confederate faced the corner, the experimenter went to the table and moved the ring from one box (henceforth referred to as the now-empty box) to the other box (henceforth the now-full box) and then shut both boxes by pushing down on both lids. Then, the experimenter left the room using a door situated next to the corner that the confederate was facing (figure 1). After about 1000 ms, the confederate closed the door, turned around, walked towards the table and stepped back on the mWBB. For approximately 500 ms, the confederate maintained a gaze equidistant between the two boxes. Then the confederate oriented her gaze first towards the now-full box and then towards the now-empty box, and finally reached to open the latter. After about 2000 ms of unsuccessfully trying to open the now-empty box, the confederate assumed a neutral but natural position on the mWBB and asked the participant: 'Can you please help me?' After the participant had helped to open a box, the confederate called the experimenter back into the room and the session ended.

For participants in the TB condition, the experimenter spoke aloud to the confederate as follows. 'Now I am going to ask you to put the headphones on. This is the sound you will be hearing. I will then leave the room and, as soon as you are ready, you have to close the door behind me and then you have to retrieve your object. Ok, now put the headphones on'. At this point, the experimenter went to the table and (watched by the confederate) he proceeded to move the ring from one box (the now-empty box) to the other one (the now-full box). Then the experimenter shut both boxes by pushing down both lids, and left the room. After about 1000 ms, the confederate stepped off the board, closed the door, turned around, walked towards the table and stepped back the mWBB. Then, after about 500 ms, the confederate oriented her gaze first towards the now-full box and then towards the now-empty box, and finally reached to open the latter. After about 2000 ms of

unsuccessfully trying to open the now-empty box, the confederate assumed a neutral but natural position on the mWBB and asked the participant: 'Can you please help me?' After the participant had helped to open a box, the confederate called the experimenter back into the room and the session ended.

At the end of the experimental session, participants were invited to complete an exit questionnaire. There were 4 two-alternative forced-choice items providing checks on participants' attention and memory. Each item showed a picture of the experimental table and boxes, and participants had to answer to the following questions: (i) 'Circle the box that the other participant placed her item into'; (ii) 'Circle the box that the experimenter moved the item into'; (iii) 'Circle the box that the other participant tried to open'; (iv) 'Circle the box you went to open'. If participants incorrectly answered one or more of the forced-choice attention and memory check items, their data would be excluded from formal analysis; the responses from three participants were excluded on this basis. The exit questionnaire also included two open-ended items probing for the reasons behind participants' ultimate choice of helping action: 'Why did you go for that box?' and 'Why didn't you go for the alternative box?' A final open-ended question probed whether participants happened to be familiar with or latched onto what the task was measuring by asking: 'What is the experiment about?' None of the participants indicated familiarity with the task or its purpose; all participants merely indicated that the experiment was about action observation.

## 2.3. Data analysis

We measured each participants' gazing (fixation duration to either box, first fixated box) individually during a specific time of interest, beginning after the confederate closed the door and ending before she oriented her gaze towards one or the other box (figure 2). Raw fixation durations were then transformed into proportions. To extract gaze data, we defined two same-sized ($277 \times 317$ pixels) areas of interest (AOI, figure 3)—the now-full box AOI and the now-empty box AOI—and applied the standard attention filter of the Tobii Pro Lab programme. In addition, we checked that participants' eye gaze was not followed by head/torso movements while fixating to one box or the other; this was done with the aim of excluding the possibility that participants' leaning could have been influenced by attention orientation.

We measured participants' average leaning on the WBB during a specific time of interest, beginning after the confederate closed the door and ending just before she stepped on the mWBB (figure 2). The time window for measuring participants' leaning ended just before the confederate stepped on the mWBB to avoid confounding effects that the confederate's action of stepping on the balance board (e.g. she raises one leg, she sways to restore the balance, she places her foot and so on) could have had on observers' own motor system. This time window was fixed for all participants and lasted 2020 ms; the trained confederate never took less than 2020 ms to go from the door to the balance board. The eye-gaze time window, however, included the moment when the confederate was on the mWBB and ended just before she oriented her gaze towards a box, to give us the best chance of detecting first fixations. In contrast with the consistent output from the balance board, some gaze-signal loss is inevitable due to the nature of eye tracking technology, and consequently, the eye-gaze time window was coded individually for each participant (and individuals' raw data were then transformed into proportions).

The WBB has been proven to be a reliable tool providing temporally and spatially sensible information about the body's centre of pressure (COP) [15,21,31]. The COP is defined as the orthogonal projection of the centre of gravity on a horizontal plane, and the WBB expresses its left–right position on the $x$-axis in centimetres. We were able to investigate mediolateral shifts in balance—posture exhibiting a lean towards the now-full box or towards the now-empty box—during the time of interest by calculating the average participants' COP displacement from the COP position at the beginning of the time of interest. Finally, we also coded which box participants ultimately chose to help with (either by opening or touching). For statistical analysis, IBM SPSS Statistics 24 was used. The significance level for all analysis was $p \leq 0.05$, two-tailed.

# 3. Results

## 3.1. Gaze analyses

A mixed analysis of variance with box (now-full, now-empty) as within-subjects factor and condition (FB, TB) as between-subjects factor was performed on the proportion of duration (in percentage) that

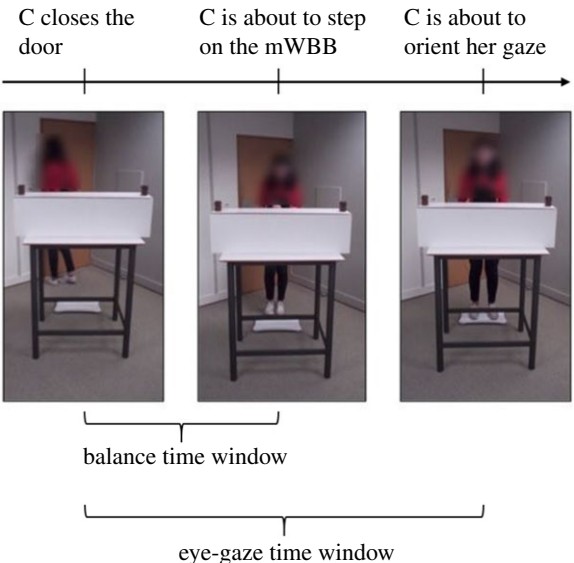

C closes the door | C is about to step on the mWBB | C is about to orient her gaze

balance time window

eye-gaze time window

**Figure 2.** Schematic of the selected time windows. The balance time window had a fixed duration of 2020 ms, while the eye-gaze time window was selected individually for each participant (raw data were then transformed into proportions). The face of the confederate agent is blurred only for the purposes of publication.

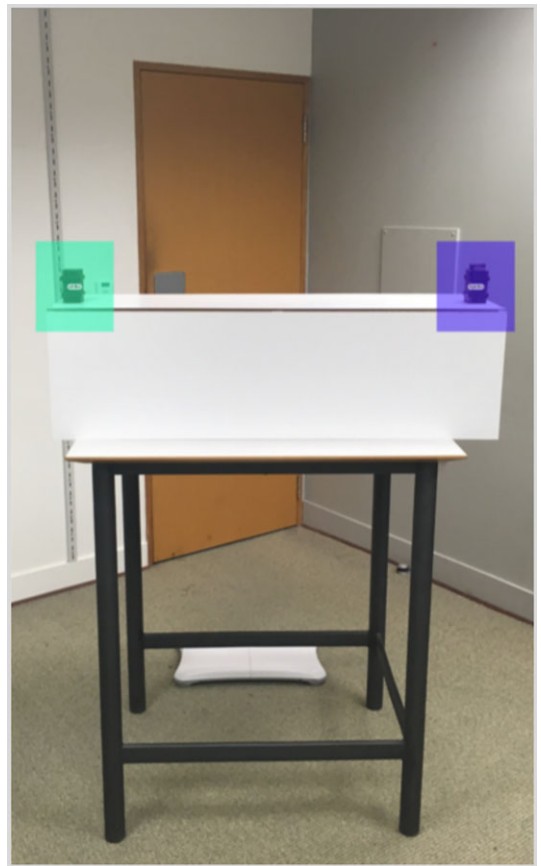

**Figure 3.** Two same-size AOI (277 × 317 pixels) were selected for eye-gaze data analysis.

participants ($N = 42$) spent fixating to either box. The results showed a significant interaction between box and condition ($F_{1,40} = 8.674$, $p = 0.005$, $\eta_p^2 = 0.178$). Specifically, *post hoc* comparisons revealed that participants in the TB condition fixated longer ($p = 0.022$, $\eta_p^2 = 0.125$) to the now-full box ($M = 46.247$; s.e. = 8.215; 95% CI: 29.643–62.852) compared to the now-empty box ($M = 16.253$; s.e. = 8.018; 95%

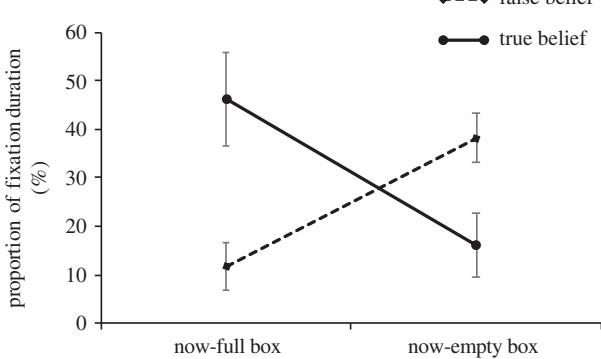

**Figure 4.** Fixation duration (in %) on each box (now-full, now-empty) in FB and TB conditions.

CI: 0.048–32.457) and that the now-full box was fixated longer ($p = 0.009$, $\eta_p^2 = 0.159$) by participants in the TB condition ($M = 46.247$; s.e. $= 8.215$; 95% CI: 29.643–62.852) compared to participants in the FB condition ($M = 11.715$; s.e. $= 9.486$; 95% CI: $-7.458$ to 30.888) (figure 4).

With respect to first looks, 24 participants (57%) showed a clear first fixation to either the now-empty box or the now-full box in the specific time window of interest; all of these participants were gazing at the confederate's hand at the beginning of the critical time window, so their first fixations were not the result of them already looking at one or the other box. The remaining 18 participants (43%) did not show any first looks to either box during the specific time window, either because they looked anywhere outside the selected AOIs (11 participants) or because the eye-movements signal was lost (7 participants). Among the 24 participants who did show a first fixation, no significant effects emerged from the Fisher exact test to determine whether there was any relationship between condition (FB, TB) and the box (now-full box, now-empty box) that was first fixated upon (Fisher exact test, $p = 0.241$) (figure 5).

## 3.2. Balance analyses

We next examined participants' mediolateral leaning on the WBB ($N = 37/42$; balance board data of four participants were not acquired due to technical problems; balance board data of one participant were excluded for exceeding the mean more than 2 s.d.). A Mann–Whitney $U$-test was conducted to determine whether there was a difference in the average leaning between those assigned to the FB condition and those assigned to the TB condition. Results revealed a significant group difference ($z = -2.486$, $p = 0.012$, $\eta^2 = 0.193$; figure 6), with participants in the FB condition leaning towards the now-empty box ($M = -0.09$, s.d. $= 0.21$, CI $= -0.56$, 0.18) and participants in the TB condition leaning towards the now-full box ($M = 0.07$, s.d. $= 0.14$, CI $= -0.25$, 0.30).

## 3.3. Final helping action analyses

A $\chi^2$ test, performed on which box participants chose to help with (either by opening or touching), showed that final helping action was not significantly different between conditions ($\chi^2(1, N = 42) = 0.146$, $p = 0.703$, figure 7). In general, most participants ($N = 36/42$, 86%) decided to help with the box that the confederate agent was struggling to open (i.e. the now-empty box in the FB and in the TB conditions, 83% and 87%, respectively). The percentages of participants selecting to help open the now-empty box in the FB and TB conditions were different from chance by a binomial test ($p = 0.003$ with Cohen's $g = 0.33$, and $p = 0.001$ with Cohen's $g = 0.37$, respectively). Further, no significant effects emerged from a Fisher exact test that we ran to investigate whether there was any relation between the first fixated box and the box participants ultimately chose to help with (Fisher exact test, $p = 0.634$).

We analysed the reasons participants offered for their final helping action. Following a common practice in theory-of-mind studies that work with narrative-based or categorical responses [32–34], two raters independently coded 24% ($n = 10$) of participants' reasons for their final helping action, given to question #1 (Explain why you went to that box) and question #2 (Explain why you didn't go to the alternative box). Both raters were blind to the condition (TB or FB) that the participants' answers belonged to. The raters independently coded the explanations into one of three categories. One category picked up on matters of fact (e.g. 'Because that was [the box] that wasn't being used'; 'It was the box that the item was originally placed in'). Another category picked up on explanations

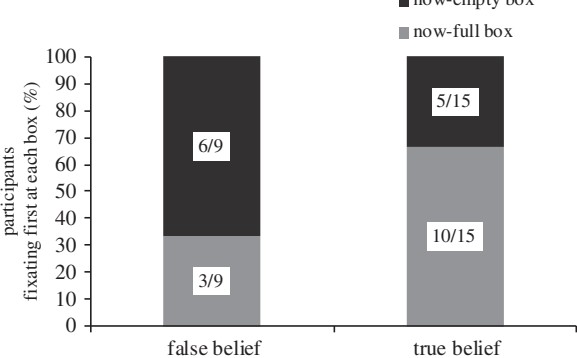

**Figure 5.** Number of participants displaying a first fixation either to the now-empty box or now-full box during the critical time window ($N = 24$) by condition (FB or TB).

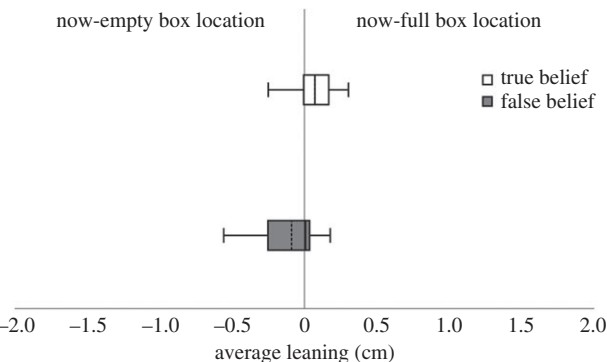

**Figure 6.** Graphical representation of displacement from body midline (0) split for group (FB, TB). Positive values reflect a leaning towards the box with the object in it (now-full box); negative values reflect a leaning towards the box in which the object was initially located (now-empty box). Dotted line in the box represents the mean, continuous line represents the median, length of the box represents the interquartile range and the whiskers extend to the highest and lowest observations.

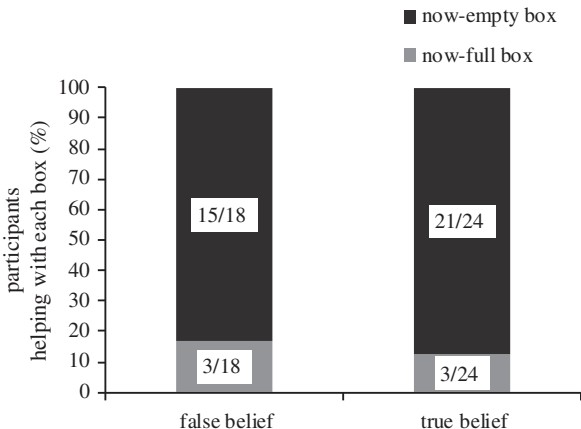

**Figure 7.** Number of participants who helped with either the now-empty or the now-full box ($N = 42$) by condition (FB or TB).

that referred to someone's general ability to try to open a box (e.g. 'This was the one she was struggling with'; 'Because it was the other box that the person was having trouble with'). The final category picked up on any mental states relating to desires, knowledge, perceptions or thoughts (e.g. 'Because I need to help get the item and I know it was in that box'; 'Because I was helping the person open the box they thought the item was in'). The average of the two raters' classifications of the participants' explanations produced an inter-correlation coefficient of 0.91, $p < 0.001$, which represented excellent reliability. One of the raters then coded the remaining explanations. The results were straightforward (table 1). Most participants in the TB condition (83%, $n = 19/23$) and FB condition (72%, $n = 13/18$)

**Table 1.** Numbers of participants in the TB and FB conditions ($n = 23$ and $n = 18$, respectively) who provided fact-based, ability-based or mental-state-based reasons for the questions that sought explanations of why they went to open one box and did not go to open the alternative box. One participant in the TB condition did not answer these questions, and hence the coding of reasons in TB condition was based on data from 23 participants.

| reason | those who opened now-empty box | | | those who opened now-full box | | |
| --- | --- | --- | --- | --- | --- | --- |
| | fact | ability | mental state | fact | ability | mental state |
| *Question 1: Explain why you went to that box* | | | | | | |
| TB | 8 | 11 | 1 | 0 | 0 | 3 |
| FB | 2 | 11 | 2 | 1 | 0 | 2 |
| *Question 2: Explain why you didn't go to the alternative box* | | | | | | |
| TB | 10 | 5 | 5 | 1 | 0 | 2 |
| FB | 4 | 4 | 7 | 2 | 0 | 1 |

referred to facts or the agent's general ability to open a box when explaining why they went to help open the now-empty box (e.g. 'They were struggling with that box, so I showed them how to do it'). Many participants in the TB condition who opened the now-empty box (65%, 15/23) also talked about facts or the agent's general ability when explaining why they did not open the alternative (now-full) box (e.g. 'It felt more natural to walk over to where the person was standing'; 'Because the person was not trying to open that one'). With respect to those participants opening the now-empty box in the FB condition, the participants provided fact- or ability-based explanations (53%, 8/15) or mental-state explanations (47%, 7/15) for not opening the alternative (now-full) box. Those mental-state explanations could suggest that some adults may also not necessarily use their ability to rapidly track belief for personally correcting an agent's false-belief during the final helping action.

## 4. Discussion

The present research confirmed adult observers' FB tracking ability as manifested in certain anticipatory gaze response patterns. In the FB condition, just before the agent was about to select an action, participants looked in anticipation towards the empty box. Correspondingly, in the TB condition, just before the agent was about to select an action, participants looked in anticipation towards the full box. These results dovetail with those studies showing that specific eye gazing can reveal adults' ability to quickly and correctly anticipate the action of a person who has an FB or a TB about the location of an object [5,19].

Our study also confirmed whether the pattern of eye gaze responding was replicated in observers' spontaneous motor representations of action prediction. Previous research has studied motor representation of action prediction mainly in relatively simple tasks where the outcome to which an action is directed is going to be successfully achieved by the agent. In our interactive task scenario, however, the agent has an FB of an object's whereabouts as well as which action is needed. Fixing only upon the target of the agent's goal (the particular outcome that that her bodily movements are directed towards, like reaching and grasping the ring) would lead participants to generate the wrong prediction about how the agent's course of action would unfold.

Our analyses of adults' mediolateral balance shifts—sampled at an early time point in the event sequence when there were no overt cues to suggest which box the agent would move towards— confirmed spontaneous motor representation of belief-based actions by the observers in response to the agent's predicament. In the FB condition, adults leaned towards the empty box; and in the TB condition, they leaned towards the full box. These results document, for the first time, adults' motorically grounded expectations of the agent's action being modulated by the workings of an FB tracking system. These results suggest that motor representations and processes can go beyond mere goal ascription; they can successfully accommodate cases where belief-tracking informs goal ascription.

Adjustments in adult observers' own mediolateral leaning occurred before the agent even performed any overt reaching movement towards a particular box location, as if observers' motor activity anticipated the likely target of the agent's upcoming belief-based action. The fact that adults leaned to

their own right side in anticipating that the agent would—from her perspective—go to the left-side box, and that adults leaned to their own left side when anticipating that the agent would—from her perspective—go to the right-side box, fits with computational and conceptual models suggesting that motor representations and processes may be able to remap the agent's allocentric frame of reference into subjects' own egocentric frame of reference [35]. There are, however, studies showing that the link between action observation/prediction and action execution can be motorically mapped in some somatotopic manner (e.g. see [36] for a review). For example, adults observing a needle penetrating the hand of a human model showed changes in corticospinal motor representations in the particular muscle that was pricked [37]. Adults also show action priming effects when congruent body effectors are involved [38]. Nonetheless, there is also more to the dynamics of motor representations and processes. Many studies show that there is selective discharge in motor activity according to the goal an action is directed towards, regardless of the specific effector used [7,39]. For example, adults observing someone wearing a miniaturized soccer shoe kicking a ball with the index finger showed motor facilitation in their leg [40]. There is broader evidence, then, that lends weight to our findings, suggesting that motor representation and processing of another's action towards an object is not just a matter of muscle, effector or posture specific resonance but, more importantly, of the belief-informed goal another's action is directed to.

If connections between motor and mindreading processes allow us to understand situations where belief-tracking informs goal ascription, then this is useful for social cognition. But how should we characterize the connections? Research on motor processing suggests that an onlooker's motor system generates expectations taking into account facts about the actual environment (e.g. barriers that might block someone's possibility to act in reaching space) [13]. Such motor processes occur not only when a subject is observing an actual environment but also when she or he is imagining it [14]. Butterfill & Apperly's [1] novel suggestion is that, during action observation, the onlooker's motor system is not tied to the actual environment but can also generate behavioural expectations based on non-actual environments and, in particular, expectations which track the agent's beliefs. Our findings could be seen as supporting this suggestion. Alternatively, it may be that the belief-tracking process guides subjects in spontaneously predicting the agent's future actions non-motorically, and these predictions then trigger motor activity. On the alternative view, belief-tracking and motor processes would not be integrated in the way that Butterfill and Apperly envisage. In order to decide between these competing ideas, further research is necessary. For instance, if motor processing and belief-tracking are tightly integrated, temporarily impairing one should have an effect on the other. Thus, with respect to our task scenario, impairing subjects' abilities to represent actions motorically (perhaps by using bodily constraints [41] or transcranial magnetic stimulation [42]) would not only disrupt participants' mediolateral leaning (cue to motor processing of expected actions) but would correspondingly disrupt anticipatory gazing (cue to FB tracking process). On the other hand, if what the motor system gets as input from the FB tracking process is some analysed visual information about the another's belief-like state (on the agent, the object and spatial relation between them) then impairing subjects' abilities to represent actions motorically might selectively disrupt participants' mediolateral leaning but spare anticipatory gazing.

Since the use of the WBB in conjunction with a mobile eye tracker for answering questions about belief-based action understanding is novel, and since effect sizes are not particularly large, we should also be cautious in the interpretations of our work. First, the naturalistic setting of our task meant that we cannot rule out the possibility that unconscious changes in the confederate's behaviour (such as the confederate's kinematics, locomotion or breathing) may have influenced the early indicators of participants' action understanding. Future research could minimize the possibility of unconscious priming by measuring participants' mediolateral leaning and anticipatory gazing in an onscreen experiment to see whether similar effects can be obtained without live action. Second, the naturalistic setting of our task meant that we were not able to incorporate 'attention-getters' into the scene, which would have otherwise ensured that participants' gaze could be reliably oriented to a point equidistant from the target locations before the measurement phase. Future research would do well in administering computer-based versions of our task that also controls for gaze location prior to the measurement onset. Such computer-based applications of our work could then pave the way for exploring finer-grained links between balance and gaze data (e.g. whether individual differences in gaze pattern might be predictive of different leaning tendencies). Overall, though, the converging evidence of balance and gaze measures provides empirical footing for a connection between fast tracking of others' beliefs and motor processes, leaving room for further research to pin down the extent of such a connection.

Adults' final helping actions were unlike those which have been reported for young children in a similar helping task [16]. Adults (different from young children) were not more likely to help open

the now-full box in the FB condition than in the TB condition. The majority of adults helped to open the now-empty box in both conditions. When we asked them why they chose this box, adults generally referred to facts relating to the situation or to the agent's abilities, rather than to mental states *per se*.

Why did the final helping response not reflect belief information that adults' eye gaze and leaning responses picked up upon? One possibility is that adults had the agent's mental state in mind all along but took the view it was normatively irrelevant to their choice of action. This possibility is consistent with the observation that no adults expressed the view that their choice of final helping action was mistaken when subsequently asked why they had acted as they did. Alternatively, it may be that adults are poor at reflecting on the agent's belief in the course of socially interacting with her, despite the occurrence of FB tracking. Much as adults have been shown to not always put to use another's visual perspective when interacting with her [43], so also they may cut corners by ignoring another's belief. Importantly, our results indicate that they could be doing this even when their eye gaze and leaning indicate that they have tracked the belief: if so, there may be a dissociation in adults' performance analogous to that sometimes observed in young children [44–46]. Future research will need to distinguish between these different reasons for why, despite showing sensitivity to belief in eye gaze and leaning, the adults did not do so in their final helping actions.

Why might adults' final helping responses be different from children's, at least in this respect? One possibility is that adults and children have different views about what one should do. Adults may generally think it is best to do as expected, whereas children may be more disposed to act in another's best interests. Alternatively, it may be that children's actions are less reflective than adults. The FB tracking processes which underpin adults' eye gaze and modulate adults' leaning also occur in children but dominate children's actions for longer; in adults, those FB tracking processes are overshadowed by a more reflective process which, however, can be slow to incorporate information about beliefs (and other mental states, and perspective) in reaching decisions about what to do, especially when they are involved in a social interaction with the agent. Yet, another way to think about the pattern of adults' final helping responses is to consider the possibility that—unlike in infancy studies on theory of mind where subjects are repeatedly familiarized with the agent having the goal of retrieving the hidden object—the confederate's goal could only be indirectly inferred by appealing to the fact that the hidden object was a confederate's belonging. It is possible that adults' final belief-based helping responses may be elicited under conditions where the box-opening action is even more clearly linked to the goal of retrieving the confederate's possession. Regardless of the specific reasons for adults' own choice of final helping response, the findings highlight the fact that new questions and answers into the developmental processes underpinning tracking and using of belief information can be broached and gained by studying action understanding in mature populations [47].

Our findings also provided us with two unexpected payoffs. First, some failed replications of the finding that anticipatory looking can reflect FB tracking have led to the suggestion that eye gaze measures may be altogether ill-suited to measuring action anticipation [48]. To get clearer about the validity of eye gaze as reflecting belief-tracking, it is worthwhile considering that, to date, anticipatory looking behaviour towards locations of belief attribution has been demonstrated often in relatively passive computer-based tasks (of course, as discussed here, our interactive task too has complementary design drawbacks). This restriction of computer-based tasks invites the charge that apparent FB effects in eye gaze are merely artefacts which do not reflect abilities important for everyday social interactions. On the other hand, if eye gaze can really be used to measure belief tracking, it should be possible to find real-time situations in which subjects' eyes reveal what they take another to believe [49]. Our results extend documentation of rapid mental-state tracking to a real-time interactive helping scenario. This gives weight to the idea that eye gaze is a valid cue to belief-tracking [50], and supports claims that human beings' FB tracking is a vital social sense for the monitoring of others' action preparations [2,51]. Second, if FB tracking can only affect our eyes in tasks that involve relatively passive looking expectations, for example, its function will remain mysterious. Since our own physical movements also foreshadow other people's belief-like state, our findings would suggest that the FB tracking system, by way of its connections with the motor system, can have functional consequences for action understanding.

# 5. Conclusion

We measured adults' belief-tracking and motor processing in an interactive helping task. Prior to an agent's selection of a particular course of action, participants' own action predictions—as reflected in anticipatory gazing as well as mediolateral leaning—foreshadowed the agent's FB or TB of an object's

location. Our findings suggest that fast tracking of others' beliefs interfaces with motor processes, to the extent that it can influence action prediction and set our bodies swaying. We hope this work can inspire more studies that will advance our understanding of how adults' FB tracking variously interface with, and perhaps even depend on, motor processes.

Ethics. The School of Psychology Human Ethics Committee under delegated authority from Central Human Ethics Committee of Victoria University of Wellington provided ethics approval for the study, and informed consent was obtained from all individual participants included in the study.

Data accessibility. The datasets supporting this article have been uploaded as part of the electronic supplementary material.

Authors' contributions. G.Z. conceived and designed the study, collected, coded, analysed and interpreted the data, and drafted the article. S.A.B. and J.L. advised on conception and design of the study, helped to interpret the data, and helped to edit and revise the article to clarify important intellectual content. All authors gave final approval for publication.

Competing interests. We have no competing interests.

Funding. This research was funded by the Royal Society of New Zealand Marsden Fund (17-VUW-074, contract VUW1706).

Acknowledgements. We thank Katheryn Edwards, Sumaya Lamb and Pieter Six for their research assistance.

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
