## [Reviewer comments · Royal Society Open Science]

Review History

RSOS-191167.R0 (Original submission)

Review form: Reviewer 1

Is the manuscript scientifically sound in its present form?

Yes

Are the interpretations and conclusions justified by the results?

Yes

Is the language acceptable?

Yes

Do you have any ethical concerns with this paper?

Yes

Have you any concerns about statistical analyses in this paper?

No

Recommendation?

Accept with minor revision (please list in comments)

Comments to the Author(s)

I read this manuscript with great interest as it has a very close connection to my own research. The paper reports an interesting study on the link between mediolateral motor control and mindreading. Precisely, the study reports that leftwards or rightwards leaning of participants on a balance board might reveal their access to another agent's belief, in a helping interactive task. Overall, I found the method innovative and enjoyed reading the manuscript. Despite this global positive evaluation, I have some concerns with the manuscript. In order of appearance:

- 1) Even if it is a behavioral experiment that it reported here, the authors tend to flirt with neuroimaging works. I am okay with that but I was then surprised that there was no mention of the famous study from Umiltà et al., (2001, Neuron) which address a question directly linked to the current introduction.
- 2) When referring to "G*Power 3.1", I thought that the authors of the current manuscript may want to acknowledge the authors of the software by citing the referred article.
- 3) Were the data excluded (16% of participants) well distributed among the two conditions?
- 4) I don't understand why the authors selected different time windows for the eye-gaze data and balance data analyses. Particularly, why they selected a shorter time window for the balance is not clear from my point of view. The motivation for having implemented these different time windows should be detailed in the manuscript, or, the same time window should be used for both analyses.
- 5) In figure 6, I am confused about the 4 points qualified as "outliers" by the authors. As I understand it, there is only one outlier (that has been excluded for exceeding the mean more than 2 standard deviations). If the "points" are actually part of the data being analyzed, I would not qualify them as outliers but rather as the highest and lowest observations in each condition (which are supposed to be represented by the whiskers?!). I guess that the whiskers should probably be extended to these "points".
- 6) Is there any link between the gaze data and the balance data? In the False beliefs condition for example, we can see that half of the participants did not lean toward the now-empty box (figure 6). Could different gaze pattern among participants explain the leaning tendencies?
- 7) In the current form, I'm not convinced that the leaning tendencies reported would reflect any "observers' motor activity that would mirror another agent's future belief-based action". It could just be the consequences of attention orientation toward the object of interest for the other agent (note that I am not contesting the fact that participants represent the confederate's beliefs, I am simply not convinced by the idea that the leaning would reveal the mirroring of a future action).
- 8) I might have missed something but I don't understand why the two raters only coded 24% of participants' reasons for their final helping action. Why didn't they code all participants' reasons? How were coded the other reasons? This is not clear to me...

Review form: Reviewer 2**Is the manuscript scientifically sound in its present form?**

Yes

Are the interpretations and conclusions justified by the results?

Yes

Is the language acceptable?

Yes

Do you have any ethical concerns with this paper?

No

Have you any concerns about statistical analyses in this paper?

Yes

Recommendation?

Major revision is needed (please make suggestions in comments)

Comments to the Author(s)

The study focuses on the possible interaction between false beliefs tracking and motor processes in action anticipation. The research question is extremely relevant for our theoretical development around spontaneous/implicit mentalizing. The methodological approach adopted (i.e., combining eye tracking and balance analysis) is certainly the best I have seen so far to tackle this question.

I have however a major point concerning experimental procedure and data analysis. How exactly event timing within a trial was defined and controlled? This is very relevant, especially for data analysis. At page 6, the authors wrote: "After about 1000 ms, the confederate closed the door...for approximately 500 ms, the confederate maintained a gaze equidistant between the two boxes." How were these time windows identified? I understand the confederate was instructed on what to do and when. I doubt, however, that this sequence of actions could be performed every time with such a temporal precision and speed. Were all trials video-recorded and were the videos coded off-line by an independent observer? Was the exact timing of events coded separately for each participant and the individual timing used for single subject analyses?

Page 11, line 7. "There are, however, studies showing that the link between action observation/prediction and action execution can be motorically mapped in some somatotopic manner." The authors can definitively refer here to more work (e.g., see Fadiga et al., 2005 for a review). Fadiga, L., Craighero, L., & Olivier, E. (2005). Human motor cortex excitability during the perception of other's action. *Current Opinion in Neurobiology*, 15, 213-218.

Review form: Reviewer 3 (Denis Tatone)

Is the manuscript scientifically sound in its present form?

Yes

Are the interpretations and conclusions justified by the results?

Yes

Is the language acceptable?

Yes

Do you have any ethical concerns with this paper?

No

Have you any concerns about statistical analyses in this paper?

No

Recommendation?

Accept with minor revision (please list in comments)

Comments to the Author(s)

The paper is well written and well argued. The literature was adequately surveyed and the hypotheses clearly stated. I applaud the authors for having substantiated with cogent arguments their determination of sample size (as per recommendations of Simmons et al. 2012), and for having ingeniously complemented established measures of belief-tracking with new ones. The converging evidence of balance and gaze measures, notwithstanding issues with the first-fixation measure (see below), provides empirical footing for the claim that fast tracking of others' beliefs interfaces with motor processes, leaving room for further research to explore the extent of such interfacing.

I have no major comments for the study. Minor comments can be found below.

The confederate was not blind to condition (she could not have been, due to design constraints). This would not be a cause of concern, had no action been performed during the critical measurement window. However, in the study the confederate closed the door and placed herself on the mWBB, all during the balance and eye-gaze window. This raises the possibility of unconscious priming: subtle postural or locomotory changes in the confederate behavior during the critical testing window may have driven the participants' attention to one of the two boxes conformingly to the authors' desired outcomes. The participants' swaying and looking towards one of the two boxes may have been influenced by directional cues in the confederate's behavior, rather than by her beliefs. I am dubious that clever-Hans effects may account for the present results. However, given the authors' (justified) highlighting of the merits of their paradigm over passive computer-based ones, it is worth reminding that the former too has its design drawbacks;

I struggled with understanding how a subset of participants could not have produced first looks to one of the boxes, given that the only AOIs considered each contained a box. The authors may wish to expand on how first looks were determined. It was also unclear to me how we can reasonably interpret first looks, given that the participants' gaze could not be reliably oriented to a point equidistant from the target locations before the measurement phase (given the absence of attention-getters, a standard feature of computer-based versions of such tasks). This makes the measurement of first looks particularly noisy, in light of the inability to control for gaze location prior to the measurement onset;

The failure to replicate Buttelmann's findings is illuminating, and even more so the evidence that participants intervened on the box that the confederate attempted to open, independently of her belief about its content. From a pragmatics standpoint, these findings can be accounted for by appealing to strength-of-evidence arguments (below), which the authors did not consider in their otherwise detailed discussion on this. Unlike in infancy studies on ToM, where subjects are repeatedly familiarized with the confederate having the goal of retrieving the hidden object, no such evidence was given in this study (the goal could only be indirectly inferred by appealing to the fact that the hidden object was a confederate's belonging). Absent such prior, the participants had no strong reason to assume that the confederate's intervention on the box was aimed at retrieving the object. Without such reason, no evidence weighted against the possibility that the confederate may have goals other than that of retrieving the object for trying to open one box. This account, orthogonal to the ones fleshed out by the authors, predicts that belief-tracking helping may be elicited under conditions where the box-opening action is unambiguously linked to the goal of retrieving the confederate's possession;

Effect sizes for post-hoc analyses should be reported;

Looking time averages should be fitted with measurement units, where missing (s);
The duration of each measurement window in Figure 2 should be reported.

Review form: Reviewer 4

Is the manuscript scientifically sound in its present form?

Yes

Are the interpretations and conclusions justified by the results?

No

Is the language acceptable?

Yes

Do you have any ethical concerns with this paper?

No

Have you any concerns about statistical analyses in this paper?

Yes

Recommendation?

Major revision is needed (please make suggestions in comments)

Comments to the Author(s)

I found this manuscript clear and well-written, and I do believe that the question being explored is a valuable contribution to the literature on action understanding and belief-tracking. I did have a few questions / concerns about the paper, which I will detail below.

- The main concern I had, was about the precise time windows chosen for the eye-tracking and postural leaning analyses. It did not become clear to me why the authors had chosen a shorter time window for the latter (only until 'confederate about to step on WBB', as compared to 'confederate about to orient'). In neither case has the confederate already given any cue as to which action she is going to perform. Also, I wondered where the 2020 ms came from (as I am guessing timings may have varied between individual sessions). Have the authors also analysed data for the full window for postural leaning, or for the shorter window for eye-gazing? Are results similar?

- In line with this, upon inspecting the raw data, and seeing that the confidence intervals in Fig 6 overlap to a great extent, I think the results regarding postural leaning should be interpreted with more caution. In fact, in the False Belief condition, only 9 out of 18 participants lean more towards the 'now-empty box', and the mean difference may mainly be driven by 1 participant leaning considerably more in that direction (-1.4). Since the use of these balance boards for answering questions in the field of action understanding is novel, and since differences are not particularly large, I would apply more caution in the interpretation.

- It would be good if the authors could mention effect sizes in addition to p-values.

- In section 3.1, it should already be mentioned how the final 42 participants are divided over the FB and TB conditions (as sample sizes are not equal).

- I was not entirely convinced that participants were completely unaware of the experiment goal, as this assumption is based solely on their answer to the question 'what was the experiment

about?' Is there any more evidence to show that participants did not realise the other person in the room was a confederate (despite not wearing the Tobii glasses)? In any case, I would recommend a question asking specifically this for any future studies, as this may have influenced participants' helping behaviours.

- Finally, I would like to point the authors to a recent paper that I believe is very relevant to their introduction + discussion: Thompson, Bird, & Catmur (2019): Conceptualizing and testing action understanding, *Neuroscience & Biobehavioral Reviews*.

Decision letter (RSOS-191167.R0)

04-Nov-2019

Dear Mr Zani,

The editors assigned to your paper ("Mindreading in the balance: Adults' mediolateral leaning & anticipatory looking foretell others' action preparation in a false-belief interactive task") have now received comments from reviewers. We would like you to revise your paper in accordance with the referee and Associate Editor suggestions which can be found below (not including confidential reports to the Editor). Please note this decision does not guarantee eventual acceptance.

Please submit a copy of your revised paper before 27-Nov-2019. Please note that the revision deadline will expire at 00.00am on this date. If we do not hear from you within this time then it will be assumed that the paper has been withdrawn. In exceptional circumstances, extensions may be possible if agreed with the Editorial Office in advance. We do not allow multiple rounds of revision so we urge you to make every effort to fully address all of the comments at this stage. If deemed necessary by the Editors, your manuscript will be sent back to one or more of the original reviewers for assessment. If the original reviewers are not available, we may invite new reviewers.

- Data accessibility

If you wish to submit your supporting data or code to Dryad (<http://datadryad.org/>), or modify your current submission to dryad, please use the following link:
<http://datadryad.org/submit?journalID=RSOS&manu=RSOS-191167>

- Competing interests

- Authors' contributions

- Acknowledgements

- Funding statement

Kind regards,
Andrew Dunn
Senior Publishing Editor
Royal Society Open Science
openscience@royalsociety.org

on behalf of Dr Antonia Hamilton (Associate Editor) and Essi Viding (Subject Editor)
openscience@royalsociety.org

Comments to Author:

Reviewers' Comments to Author:

Reviewer: 1

Comments to the Author(s)

I read this manuscript with great interest as it has a very close connection to my own research. The paper reports an interesting study on the link between mediolateral motor control and mindreading. Precisely, the study reports that leftwards or rightwards leaning of participants on a balance board might reveal their access to another agent's belief, in a helping interactive task. Overall, I found the method innovative and enjoyed reading the manuscript. Despite this global positive evaluation, I have some concerns with the manuscript. In order of appearance:

- 1) Even if it is a behavioral experiment that it reported here, the authors tend to flirt with neuroimaging works. I am okay with that but I was then surprised that there was no mention of the famous study from Umiltà et al., (2001, Neuron) which address a question directly linked to the current introduction.
- 2) When referring to "G*Power 3.1", I thought that the authors of the current manuscript may want to acknowledge the authors of the software by citing the referred article.
- 3) Were the data excluded (16% of participants) well distributed among the two conditions?
- 4) I don't understand why the authors selected different time windows for the eye-gaze data and balance data analyses. Particularly, why they selected a shorter time window for the balance is not clear from my point of view. The motivation for having implemented these different time windows should be detailed in the manuscript, or, the same time window should be used for both analyses.
- 5) In figure 6, I am confused about the 4 points qualified as "outliers" by the authors. As I understand it, there is only one outlier (that has been excluded for exceeding the mean more than 2 standard deviations). If the "points" are actually part of the data being analyzed, I would not qualify them as outliers but rather as the highest and lowest observations in each condition (which are supposed to be represented by the whiskers?!). I guess that the whiskers should probably be extended to these "points".
- 6) Is there any link between the gaze data and the balance data? In the False beliefs condition for example, we can see that half of the participants did not lean toward the now-empty box (figure 6). Could different gaze pattern among participants explain the leaning tendencies?
- 7) In the current form, I'm not convinced that the leaning tendencies reported would reflect any "observers' motor activity that would mirror another agent's future belief-based action". It could just be the consequences of attention orientation toward the object of interest for the other agent (note that I am not contesting the fact that participants represent the confederate's beliefs, I am simply not convinced by the idea that the leaning would reveal the mirroring of a future action).
- 8) I might have missed something but I don't understand why the two raters only coded 24% of participants' reasons for their final helping action. Why didn't they code all participants' reasons? How were coded the other reasons? This is not clear to me...

Reviewer: 2

Comments to the Author(s)

The study focuses on the possible interaction between false beliefs tracking and motor processes in action anticipation. The research question is extremely relevant for our theoretical development around spontaneous/implicit mentalizing. The methodological approach adopted (i.e., combining eye tracking and balance analysis) is certainly the best I have seen so far to tackle this question.

I have however a major point concerning experimental procedure and data analysis. How exactly event timing within a trial was defined and controlled? This is very relevant, especially for data analysis. At page 6, the authors wrote: "After about 1000 ms, the confederate closed the door...for approximately 500 ms, the confederate maintained a gaze equidistant between the two boxes." How were these time windows identified? I understand the confederate was instructed on what to do and when. I doubt, however, that this sequence of actions could be performed every time with such a temporal precision and speed. Were all trials video-recorded and were the videos coded off-line by an independent observer? Was the exact timing of events coded separately for each participant and the individual timing used for single subject analyses?

Page 11, line 7. "There are, however, studies showing that the link between action observation/prediction and action execution can be motorically mapped in some somatotopic manner." The authors can definitively refer here to more work (e.g., see Fadiga et al., 2005 for a review). Fadiga, L., Craighero, L., & Olivier, E. (2005). Human motor cortex excitability during the perception of other's action. *Current Opinion in Neurobiology*, 15, 213-218.

Reviewer: 3

Comments to the Author(s)

The paper is well written and well argued. The literature was adequately surveyed and the hypotheses clearly stated. I applaud the authors for having substantiated with cogent arguments their determination of sample size (as per recommendations of Simmons et al. 2012), and for having ingeniously complemented established measures of belief-tracking with new ones. The converging evidence of balance and gaze measures, notwithstanding issues with the first-fixation measure (see below), provides empirical footing for the claim that fast tracking of others' beliefs interfaces with motor processes, leaving room for further research to explore the extent of such interfacing.

I have no major comments for the study. Minor comments can be found below.

The confederate was not blind to condition (she could not have been, due to design constraints). This would not be a cause of concern, had no action been performed during the critical measurement window. However, in the study the confederate closed the door and placed herself on the mWBB, all during the balance and eye-gaze window. This raises the possibility of unconscious priming: subtle postural or locomotory changes in the confederate behavior during the critical testing window may have driven the participants' attention to one of the two boxes conforming to the authors' desired outcomes. The participants' swaying and looking towards one of the two boxes may have been influenced by directional cues in the confederate's behavior, rather than by her beliefs. I am dubious that clever-Hans effects may account for the present results. However, given the authors' (justified) highlighting of the merits of their paradigm over passive computer-based ones, it is worth reminding that the former too has its design drawbacks;

I struggled with understanding how a subset of participants could not have produced first looks to one of the boxes, given that the only AOIs considered each contained a box. The authors may wish to expand on how first looks were determined. It was also unclear to me how we can reasonably interpret first looks, given that the participants' gaze could not be reliably oriented to

a point equidistant from the target locations before the measurement phase (given the absence of attention-getters, a standard feature of computer-based versions of such tasks). This makes the measurement of first looks particularly noisy, in light of the inability to control for gaze location prior to the measurement onset;

The failure to replicate Butteltmann's findings is illuminating, and even more so the evidence that participants intervened on the box that the confederate attempted to open, independently of her belief about its content. From a pragmatics standpoint, these findings can be accounted for by appealing to strength-of-evidence arguments (below), which the authors did not consider in their otherwise detailed discussion on this. Unlike in infancy studies on ToM, where subjects are repeatedly familiarized with the confederate having the goal of retrieving the hidden object, no such evidence was given in this study (the goal could only be indirectly inferred by appealing to the fact that the hidden object was a confederate's belonging). Absent such prior, the participants had no strong reason to assume that the confederate's intervention on the box was aimed at retrieving the object. Without such reason, no evidence weighted against the possibility that the confederate may have goals other than that of retrieving the object for trying to open one box. This account, orthogonal to the ones fleshed out by the authors, predicts that belief-tracking helping may be elicited under conditions where the box-opening action is unambiguously linked to the goal of retrieving the confederate's possession;

Effect sizes for post-hoc analyses should be reported;

Looking time averages should be fitted with measurement units, where missing (s);
The duration of each measurement window in Figure 2 should be reported.

Reviewer: 4

Comments to the Author(s)

I found this manuscript clear and well-written, and I do believe that the question being explored is a valuable contribution to the literature on action understanding and belief-tracking. I did have a few questions / concerns about the paper, which I will detail below.

- The main concern I had, was about the precise time windows chosen for the eye-tracking and postural leaning analyses. It did not become clear to me why the authors had chosen a shorter time window for the latter (only until 'confederate about to step on WBB', as compared to 'confederate about to orient'). In neither case has the confederate already given any cue as to which action she is going to perform. Also, I wondered where the 2020 ms came from (as I am guessing timings may have varied between individual sessions). Have the authors also analysed data for the full window for postural leaning, or for the shorter window for eye-gazing? Are results similar?

- In line with this, upon inspecting the raw data, and seeing that the confidence intervals in Fig 6 overlap to a great extent, I think the results regarding postural leaning should be interpreted with more caution. In fact, in the False Belief condition, only 9 out of 18 participants lean more towards the 'now-empty box', and the mean difference may mainly be driven by 1 participant leaning considerably more in that direction (-1.4). Since the use of these balance boards for answering questions in the field of action understanding is novel, and since differences are not particularly large, I would apply more caution in the interpretation.

- It would be good if the authors could mention effect sizes in addition to p-values.

- In section 3.1, it should already be mentioned how the final 42 participants are divided over the FB and TB conditions (as sample sizes are not equal).

- I was not entirely convinced that participants were completely unaware of the experiment goal,

as this assumption is based solely on their answer to the question ‘what was the experiment about?’ Is there any more evidence to show that participants did not realise the other person in the room was a confederate (despite not wearing the Tobii glasses)? In any case, I would recommend a question asking specifically this for any future studies, as this may have influenced participants’ helping behaviours.

- Finally, I would like to point the authors to a recent paper that I believe is very relevant to their introduction + discussion: Thompson, Bird, & Catmur (2019): Conceptualizing and testing action understanding, *Neuroscience & Biobehavioral Reviews*.

Author's Response to Decision Letter for (RSOS-191167.R0)

See Appendix A.

Decision letter (RSOS-191167.R1)

02-Jan-2020

Dear Mr Zani,

It is a pleasure to accept your manuscript entitled "Mindreading in the balance: Adults’ mediolateral leaning & anticipatory looking foretell others’ action preparation in a false-belief interactive task" in its current form for publication in *Royal Society Open Science*. The comments of the reviewer(s) who reviewed your manuscript are included at the foot of this letter.

Please see the *Royal Society Publishing guidance* on how you may share your accepted author manuscript at <https://royalsociety.org/journals/ethics-policies/media-embargo/>.

Thank you for your fine contribution. On behalf of the Editors of *Royal Society Open Science*, we look forward to your continued contributions to the Journal.

Kind regards,
Andrew Dunn
Royal Society Open Science Editorial Office
Royal Society Open Science

on behalf of Dr Antonia Hamilton (Associate Editor) and Essi Viding (Subject Editor)
openscience@royalsociety.org

Appendix A

Dear Editor (and Reviewers),

Thank you for the feedback on our manuscript to RSOS. We are especially humbled by the positive consensus amongst the 4 reviewers who found significant merit in our research. We thank each of the reviewers for the clear suggestions that have helped us to improve the clarity of our manuscript. We have taken into consideration all of the reviewers' comments, and our responses to specific points are detailed below. Our responses to each of the comments, organized by reviewer, are in italics.

Sincerely yours,
Corresponding Author

Reviewers' Comments to Author:

Reviewer 1

I read this manuscript with great interest as it has a very close connection to my own research. The paper reports an interesting study on the link between mediolateral motor control and mindreading. Precisely, the study reports that leftwards or rightwards leaning of participants on a balance board might reveal their access to another agent's belief, in a helping interactive task. Overall, I found the method innovative and enjoyed reading the manuscript. Despite this global positive evaluation, I have some concerns with the manuscript. In order of appearance:

1) Even if it is a behavioral experiment that it reported here, the authors tend to flirt with neuroimaging works. I am okay with that but I was then surprised that there was no mention of the famous study from Umiltà et al., (2001, Neuron) which address a question directly linked to the current introduction.

We thank the reviewer for providing us with this suggestion. The study has been added at page 2.

2) When referring to "G*Power 3.1", I thought that the authors of the current manuscript may want to acknowledge the authors of the software by citing the referred article.

The citation has now been added.

3) Were the data excluded (16% of participants) well distributed among the two conditions?

Of those participants who were excluded from data analysis, 3 were assigned to the False Belief condition and 5 were assigned to the True Belief Condition. We have added this information to the text.

4) I don't understand why the authors selected different time windows for the eye-gaze data and balance data analyses. Particularly, why they selected a shorter time window for the balance is not clear from my point of view. The motivation for having implemented these different time windows should be detailed in the manuscript, or, the same time window should be used for both analyses.

We thank the reviewer for advising us to clarify the different time windows. We have added this information in Section 3.3 of the revised manuscript where we state: "We measured participants' average leaning on the WBB during a specific time of interest, beginning after the confederate closed the door and ending just before she stepped on the mWBB (see figure 2). The time window for measuring participants' leaning ended just before the confederate stepped on the mWBB to avoid confounding effects that the confederate's action of stepping on the balance board (e.g. she raises one leg, she sways to restore the balance, she places her foot, and so on) could have had on observers' own motor system.

This time window was fixed for all participants and lasted 2020 ms; the trained confederate never took less than 2020 ms to go from the door to the balance board. The eye-gaze time window, however, included the moment when the confederate was on the mWBB and ended just before she oriented her gaze towards a box, to give us the best chance of detecting first fixations. In contrast to the consistent output from the balance board, some gaze-signal loss is inevitable due to the nature of eye tracking technology, and consequently, the eye-gaze time window was coded individually for each participant (and individuals' raw data was then transformed into proportions)."

5) In figure 6, I am confused about the 4 points qualified as "outliers" by the authors. As I understand it, there is only one outlier (that has been excluded for exceeding the mean more than 2 standard deviations). If the "points" are actually part of the data being analyzed, I would not qualify them as outliers but rather as the highest and lowest observations in each condition (which are supposed to be represented by the whiskers?!). I guess that the whiskers should probably be extended to these "points".

The four points are indeed outliers (they exceed the mean more than 2 standard deviations) and they have been included by a mistake during data processing. With thank the reviewer for pointing this out. Data analysis has been performed after removing these 4 data points and the results are in line with the previously reported findings. Updated results and Figure 6 have been added to the main text, please refer to section 4.2. for the results.

6) Is there any link between the gaze data and the balance data? In the False beliefs condition for example, we can see than half of the participants did not lean toward the now-empty box (figure 6). Could different gaze pattern among participants explain the leaning tendencies?

We agree with the reviewer that this is an important issue, and one that we wish to address in future research. At the current form of the experiment, it might not be advisable to perform a statistical comparison between balance and gaze data for classifying participants because i) the former is a continuous measure while the latter is relatively discrete and ii) participants were free to gaze outside the AOs and, further, in contrast to the consistent output from the balance board, some gaze-signal loss is inevitable due to the nature of eye tracking technology. Consequently, a direct comparison between balance and gaze data may be unstable. That said, we take up in the Discussion that it may be worthwhile for future research to explore individual differences between balance and gaze measures.

7) In the current form, I'm not convinced that the leaning tendencies reported would reflect any "observers' motor activity that would mirror another agent's future belief-based action". It could just be the consequences of attention orientation toward the object of interest for the other agent (note that I am not contesting the fact that participants represent the confederate's beliefs, I am simply not convinced by the idea that the leaning would reveal the mirroring of a future action).

We take the reviewer's point about the ambiguity over the meaning of the word 'future' action re-written that sentence in the Discussion (Section 5) with the more plausible suggestion: "Adjustments in adult observers' own mediolateral leaning occurred before the agent even performed any overt reaching movement towards a particular box location, as if observers' motor activity anticipated the likely target of the agent's upcoming belief-based action." Incidentally, to reduce the influence of attention orientation on balance data, we checked all the video recordings taken by the Tobii eye-glasses' frontal camera and we made sure that during the selected time window participants oriented their attention by gazing at one box or the other without moving their head/torso (specified in section 3.3).

8) I might have missed something but I don't understand why the two raters only coded 24% of participants' reasons for their final helping action. Why didn't they code all participants' reasons? How were coded the other reasons? This is not clear to me.

One coding practice in theory-of-mind studies that obtain and analyze narrative-based or categorical-response data is to have raters independently code anywhere from 20% to 25% of participants' responses, and - if there is a high level of inter-rater reliability (> 0.90) - one of the coders codes the rest of the participants' responses with confidence (Fitzke, Butterfill et al., 2017; Lecce, Ceccato, Rosi, Bianco, Bottioli, & Cavallini, 2019; Low & Edwards, 2018). Our adoption of this practice from the literature is now made more clear in the text (in Section 4.3) where we state: "Following a common practice in theory-of-mind studies that work with narrative-based or categorical responses (Fitzke et al., 2017; Lecce et al., 2017; Low & Edwards, 2018), two raters independently coded"

Reviewer 2

The study focuses on the possible interaction between false beliefs tracking and motor processes in action anticipation. The research question is extremely relevant for our theoretical development around spontaneous/implicit mentalizing. The methodological approach adopted (i.e., combining eye tracking and balance analysis) is certainly the best I have seen so far to tackle this question.

1) How exactly event timing within a trial was defined and controlled? This is very relevant, especially for data analysis. At page 6, the authors wrote: "After about 1000 ms, the confederate closed the door...for approximately 500 ms, the confederate maintained a gaze equidistant between the two boxes." How were these time windows identified? I understand the confederate was instructed on what to do and when. I doubt, however, that this sequence of actions could be performed every time with such a temporal precision and speed. Were all trials video-recorded and were the videos coded off-line by an independent observer? Was the exact timing of events coded separately for each participant and the individual timing used for single subject analyses?

We thank the reviewer for advising us to clarify this aspect of the procedure. In Section 3.3, we now explain: "We measured participants' average leaning on the WBB during a specific time of interest, beginning after the confederate closed the door and ending just before she stepped on the mWBB (see figure 2). The time window for measuring participants' leaning ended just before the confederate stepped on the mWBB to avoid confounding effects that the confederate's action of stepping on the balance board (e.g. she raises one leg, she sways to restore the balance, she places her foot, and so on) could have had on observers' own motor system. This time window was fixed for all participants and lasted 2020 ms; the trained confederate never took less than 2020 ms to go from the door to the balance board. The eye-gaze time window, however, included the moment when the confederate was on the mWBB and ended just before she oriented her gaze towards a box, to give us the best chance of detecting first fixations. In contrast to the consistent output from the balance board, some gaze-signal loss is inevitable due to the nature of eye tracking technology, and consequently, the eye-gaze time window was coded individually for each participant (and individuals' raw data was then transformed into proportions)."

2) Page 11, line 7. "There are, however, studies showing that the link between action observation/prediction and action execution can be motorically mapped in some somatotopic manner." The authors can definitively refer here to more work (e.g., see Fadiga et al., 2005 for a review).
Fadiga, L., Craighero, L., & Olivier, E. (2005). Human motor cortex excitability during the perception of other's action. *Current Opinion in Neurobiology*, 15, 213–218.

We thank the reviewer for this suggestion. The citation has now been added.

Reviewer 3

The paper is well written and well argued. The literature was adequately surveyed and the hypotheses clearly stated. I applaud the authors for having substantiated with cogent arguments their determination of sample size (as per recommendations of Simmons et al. 2012), and for having ingeniously complemented established measures of belief-tracking with new ones. The converging evidence of balance and gaze measures, notwithstanding issues with the first-fixation measure (see below), provides empirical footing for the claim that fast tracking of others' beliefs interfaces with motor processes, leaving room for further research to explore the extent of such interfacing.

I have no major comments for the study. Minor comments can be found below.

1) The confederate was not blind to condition (she could not have been, due to design constraints). This would not be a cause of concern, had no action been performed during the critical measurement window. However, in the study the confederate closed the door and placed herself on the mWBB, all during the balance and eye-gaze window. This raises the possibility of unconscious priming: subtle postural or locomotory changes in the confederate behavior during the critical testing window may have driven the participants' attention to one of the two boxes conformingly to the authors' desired outcomes. The participants' swaying and looking towards one of the two boxes may have been influenced by directional cues in the confederate's behavior, rather than by her beliefs. I am dubious that clever-Hans effects may account for the present results. However, given the authors' (justified) highlighting of the merits of their paradigm over passive computer-based ones, it is worth reminding that the former too has its design drawbacks;

We humbly correct a small part of the reviewer's point when the reviewer said that "confederate ... placed herself on the mWBB ... during the balance window [analysis]". As we now make clearer in section 3.3, we explained: "The time window for measuring participants' leaning ended just before the confederate stepped on the mWBB to avoid confounding effects that the confederate's action of stepping on the balance board (e.g. she raises one leg, she sways to restore the balance, she places her foot, and so on) could have had on observers' own motor system."

We agree with the reviewer, however, that our naturalistic paradigm cannot rule out the possibility that unconscious changes in the confederate's behavior might still have influenced participants' responses towards one of the two boxes. We now mention this point in our Discussion in Section 5 where we state: "Since the use of the WBB in conjunction with a mobile eye tracker for answering questions about belief-based action understanding is novel, and since effect sizes are not particularly large, we should also be cautious in the interpretations of our work. First, the naturalistic setting of our task meant that we cannot rule out the possibility that unconscious changes in the confederate's behaviour (such as the confederate's kinematics, locomotion or breathing) may have influenced the early indicators of participants' action understanding. Future research could minimize the possibility of unconscious priming by measuring participants' mediolateral leaning and anticipatory gazing in an onscreen experiment to see whether similar effects can be obtained without live action."

2) I struggled with understanding how a subset of participants could not have produced first looks to one of the boxes, given that the only AOIs considered each contained a box. The authors may wish to expand on how first looks were determined. It was also unclear to me how we can reasonably interpret first looks, given that the participants' gaze could not be reliably oriented to a point equidistant from the target locations before the measurement phase (given the absence of attention-getters, a standard feature of computer-based versions of such tasks). This makes the measurement of first looks particularly noisy, in light of the inability to control for gaze location prior to the measurement onset.

We thank the reviewer for highlighting this question and have added to the Results to clarify these points. Participants were simply instructed to observe the scene, so they were free to gaze at any point in the

space. In particular, as we now report in the text, for 18 participants was not possible to extract a first fixation because they looked anywhere outside the selected AOIs (11 participants) or because the eye-movements signal was lost (7 participants). We agree with the reviewer on the limits of our naturalistic study in attracting participants' attention at a point equidistant from the targets, and we think that future experiments should consider using attention-getters. We managed to reduce the impact of the absence of a symbolic attention-getter by counterbalancing the initial location of the target object. It is also relevant to mention that every session was recorded (by means of the camera mounted on the Tobii eye-glasses) and that by analyzing such recordings we were able to determine that at the beginning of the critical time window all the included participants were gazing at the confederate's hand: the analyzed first fixations were not the result of participants already looking at one or the other AOI.

We also incorporated the reviewer's good point about attention-getters in our Discussion (Section 5) where we state: "Second, the naturalistic setting of our task meant that we were not able to incorporate "attention-getters" into the scene, which would have otherwise ensured that participants' gaze could be reliably oriented to a point equidistant from the target locations before the measurement phase. Future research would do well in administering computer-based versions of our task that also controls for gaze location prior to the measurement onset. Such computer-based applications of our work could then pave the way for exploring finer-grained links between balance and gaze data (e.g., whether individual differences in gaze pattern might be predictive of different leaning tendencies)."

3) The failure to replicate Buttellmann's findings is illuminating, and even more so the evidence that participants intervened on the box that the confederate attempted to open, independently of her belief about its content. From a pragmatics standpoint, these findings can be accounted for by appealing to strength-of-evidence arguments (below), which the authors did not consider in their otherwise detailed discussion on this. Unlike in infancy studies on ToM, where subjects are repeatedly familiarized with the confederate having the goal of retrieving the hidden object, no such evidence was given in this study (the goal could only be indirectly inferred by appealing to the fact that the hidden object was a confederate's belonging). Absent such prior, the participants had no strong reason to assume that the confederate's intervention on the box was aimed at retrieving the object. Without such reason, no evidence weighted against the possibility that the confederate may have goals other than that of retrieving the object for trying to open one box. This account, orthogonal to the ones fleshed out by the authors, predicts that belief-tracking helping may be elicited under conditions where the box-opening action is unambiguously linked to the goal of retrieving the confederate's possession.

We agree with the reviewer that our results can be accounted by appealing to a range of evidence and we thank her/him for providing us with this explanation that we have now included in the Discussion (Section 5).

4) Effect sizes for post-hoc analyses should be reported.

We thank the reviewer for highlighting this oversight on our part. Effect sizes have now been updated in the manuscript.

5) Looking time averages should be fitted with measurement units, where missing (s).

The absolute duration of each participant's eye-gaze time window was individually measured, and Fixation Times were then transformed into proportions (visually represented in percentages in Figure 4). We have updated the vertical axis label to clarify this point.

6) The duration of each measurement window in Figure 2 should be reported.

Figure 2's label has now been updated to make more clear that the Balance Time Window had a fixed duration of 2020 ms while Eye-Gaze Time Window was individually selected for each participant.

Reviewer 4

I found this manuscript clear and well-written, and I do believe that the question being explored is a valuable contribution to the literature on action understanding and belief-tracking. I did have a few questions / concerns about the paper, which I will detail below.

1) The main concern I had, was about the precise time windows chosen for the eye-tracking and postural leaning analyses. It did not become clear to me why the authors had chosen a shorter time window for the latter (only until 'confederate about to step on WBB', as compared to 'confederate about to orient'). In neither case has the confederate already given any cue as to which action she is going to perform. Also, I wondered where the 2020 ms came from (as I am guessing timings may have varied between individual sessions). Have the authors also analysed data for the full window for postural leaning, or for the shorter window for eye-gazing? Are results similar?

We thank the reviewer for advising us to be clearer in our explanation of the time windows. These clarifications are now provided in section 3.3. We state: *"We measured participants' average leaning on the WBB during a specific time of interest, beginning after the confederate closed the door and ending just before she stepped on the mWBB (see figure 2). The time window for measuring participants' leaning ended just before the confederate stepped on the mWBB to avoid confounding effects that the confederate's action of stepping on the balance board (e.g. she raises one leg, she sways to restore the balance, she places her foot, and so on) could have had on observers' own motor system. This time window was fixed for all participants and lasted 2020 ms; the trained confederate never took less than 2020 ms to go from the door to the balance board. The eye-gaze time window, however, included the moment when the confederate was on the mWBB and ended just before she oriented her gaze towards a box, to give us the best chance of detecting first fixations. In contrast to the consistent output from the balance board, some gaze-signal loss is inevitable due to the nature of eye tracking technology, and consequently, the eye-gaze time window was coded individually for each participant (and individuals' raw data was then transformed into proportions)."*

2) In line with this, upon inspecting the raw data, and seeing that the confidence intervals in Fig 6 overlap to a great extent, I think the results regarding postural leaning should be interpreted with more caution. In fact, in the False Belief condition, only 9 out of 18 participants lean more towards the 'now-empty box', and the mean difference may mainly be driven by 1 participant leaning considerably more in that direction (-1.4). Since the use of these balance boards for answering questions in the field of action understanding is novel, and since differences are not particularly large, I would apply more caution in the interpretation.

We thank the reviewer for advising us to consider removing outliers. With respect to the one participant leaning considerably more (-1.4) towards the now empty box, his and other three data points originally labelled in figure 6 as "outliers" have now been removed because they exceeded the mean more than 2 standard deviations and it was an oversight to have them included in data analysis. Balance results and relative figure 6 have been updated in the main text. Despite these corrections, results are in line with the previously reported ones, please refer section 4.2).

We agree with the reviewer that more general caution in discussing balance data should be applied, and we have done so in the Discussion (Section 5) where we highlighted the different ways in which a cautious reading of our work is also appropriate.

3) It would be good if the authors could mention effect sizes in addition to p-values.

Effect sizes have now been updated in the manuscript.

4) In section 3.1, it should already be mentioned how the final 42 participants are divided over the FB and TB conditions (as sample sizes are not equal).

This is now specified in section 3.1.

5) I was not entirely convinced that participants were completely unaware of the experiment goal, as this assumption is based solely on their answer to the question 'what was the experiment about?' Is there any more evidence to show that participants did not realise the other person in the room was a confederate (despite not wearing the Tobii glasses)? In any case, I would recommend a question asking specifically this for any future studies, as this may have influenced participants' helping behaviours.

We agree with the reviewer that caution should be applied in making sure of the effectiveness of the deception. As a matter of fact, we were required for ethical reasons to provide a complete debriefing in which participants were explicitly told about the real role played by the confederate. After closing the deception, the experimenter asked the participant whether he/she thought that the other person was an actress and not a second participant: only one participant actually conveyed awareness about the confederate not being a second participant, and, for the sake of brevity, this participant was termed as 1 of the 3 participants that we refer as excluded due to confederate error (in section 3.1).

6) Finally, I would like to point the authors to a recent paper that I believe is very relevant to their introduction + discussion: Thompson, Bird, & Catmur (2019): Conceptualizing and testing action understanding, *Neuroscience & Biobehavioral Reviews*.

We thank the reviewer for suggesting this recent paper to us.